# AI for an inverse problem: Physical model solving quantum gravity

**Koji Hashimoto** [* 1]   **Koshiro Matsuo** [* 2]   **Masaki Murata** [* 2]   **Gakuto Ogiwara** [* 2]   **Daichi Takeda** [* 1]

## Abstract

Mathematical inverse problems of determining a governing differential equation for given solution data remain a fundamental challenge. To find a working example of AI for math, we provide a concrete example using a physical setup of a quantum gravity problem. We present a novel sparse Neural Network (NN) model which is interpretable, to solve the inverse problem: the AdS/CFT correspondence. According to the conjectured correspondence, a special condensed matter system on a ring is equivalent to a gravity system on a bulk disk. The inverse problem is to reconstruct the higher-dimensional gravity metric from the data of the condensed matter system. We use the response functions of a condensed matter system as our data, and by supervised machine learning, we successfully train the neural network which is equivalent to a scalar field equation on an emergent geometry of the bulk spacetime. The developed method may work as a ground for generic bulk reconstruction, *i.e.* a solution to the inverse problem of the AdS/CFT correspondence. From a technical perspective, to achieve better numerical control, our neural network model incorporates a novel layer that implements the Runge-Kutta method.

## 1. Introduction

Inverse problems are at the heart of mathematics, as they are indispensable for proving equivalence. In general, mathematics can provide some existence theorem which makes sure the existence of the solution of an inverse problem, but in reality for the application of the mathematical framework to sciences, the existence theorem is often of no use except

---

[*]Equal contribution  [1]Department of Physics, Kyoto University, Kyoto 606-8502, Japan [2]Faculty of Engineering of Engineering Information Technology Course Saitama Institute of Technology. Correspondence to: Koshiro Matsuo <f3011hxs@sit.ac.jp>, Masaki Murata <m.murata@sit.ac.jp>.

The first AI for MATH Workshop at the 41st International Conference on Machine Learning, Vienna, Austria. Copyright 2024 by the author(s).

for just consolation for practitioners. One major example of this kind is the universal approximation theorem of neural networks (Cybenko, 1989), which makes sure that there exists a set of neural network weights with which the output of the neural network approximates very well a given target function. As popularly known, it is a different problem to find how to update the weights to make the approximation in practice. This is related to the problem of interpretability of a given neural network.

Physical systems provide a playground for this kind of inverse problems, because one has some physical intuition for the solutions of the inverse problems. One of the major inverse problems is to find a set of differential equations which provides its solutions satisfying a certain boundary condition. One can use neural networks for solving this inverse problem of finding the governing equations, but again, the interpretability of the obtained trained neural network is in question.

In this paper, to approach this question, we set up a concrete physical model concerning the long-standing problem of quantum gravity. The most promising formulation of quantum gravity is to use the AdS/CFT correspondence (Maldacena, 1999), which is a conjecture that two physical quantum theories, conformal field theory (CFT) and gravity theory, are equivalent to each other. Unfortunately, there has been no proof of the conjecture, but there are a lot of working examples. The principal concern against this conjecture is that for a given CFT data, there is no constructive way to find a background geometry in the dual equivalent gravity theory. This is precisely the inverse problem which we stated above: CFT provides a boundary data, and the issue is to find a differential equation in a curved geometry (which is unknown) whose solution is consistent with the boundary data.

A possible solution to this inverse problem (widely known as "bulk reconstruction program" in the field of theoretical particle physics) can be a ground-breaking path to a proof of the conjecture, and furthermore, a working example for AI for math of inverse problems.

The best and the simplest setup for solving this inverse problem of the AdS/CFT correspondence is with a low-enough spacetime dimensions and the simplest matter content in the bulk spacetime, thus we follow the situation provided in

(Hashimoto et al., 2023) where the CFT (material quantum theory) lives on a one-dimensional circle while the corresponding gravity background is a disk which is rotationally symmetric. We follow the method developed in (Hashimoto et al., 2018a) which replaces the bulk differential equation with a sparse neural network securing the interpretability by regarding the weights as a metric on the curved spacetime.[1] With this setup, if we find what kind of differential equations are emergent and what are not, it would be a great step for proving the AdS/CFT correspondence — the major inverse problem in physics. And it would be a ground to develop a solving method for the inverse problem of finding consistent governing differential equations. In this paper, we would like to report our first-step results.

## 2. Physics Background

### 2.1. The AdS/CFT Correspondence (Maldacena, 1999)

The AdS/CFT correspondence (Maldacena, 1999) is a conjecture stating that a quantum gravity theory with a negative cosmological constant in $d + 1$-dimensional asymptotically AdS spacetime is equivalent to a non-gravitating quantum field theory in $d$-dimensional spacetime. The former is called bulk theory, and the latter is boundary theory. The AdS is the anti de-Sitter space, the maximally symmetric spacetime of constant negative curvature. While the conjecture has not yet been proven mathematically, various evidences are found and its application is now broad, for example to condensed matter physics and quantum computation.

Starting with a given bulk gravity theory, we can straightforwardly compute the correlators on the dual boundary CFT. All we have to do is to calculate the bulk on-shell action by solving the equation of motion (EOM) under certain boundary conditions. This on-shell action becomes the generating functional of CFT correlators of the boundary. On the other hand, it is in general hard to construct bulk fields from the boundary, as we need to identify the EOM consistent with a given set of CFT correlators. This inverse problem is called "bulk reconstruction" and a well-posed inverse problem realized in an area of physics toward quantum gravity.

### 2.2. Spacetime-emergent Material (SEM)

The spacetime-emergent material (SEM) is a ring-shaped material whose properties are equivalently described by a $(2 + 1)$-dimensional quantum gravity under the AdS/CFT

---

conjecture.

According to the AdS/CFT, when the bulk is a black hole spacetime, the corresponding boundary theory is a finite temperature CFT (Witten, 1998). Since generic condensed matter systems near a quantum critical point (QCP) are dictated by a thermal CFT, it is expected that there may exist a material which allows a higher-dimensional gravity description, which was named spacetime-emergent material in (Hashimoto et al., 2023). The simplest realization of the AdS/CFT in our world would be a ring-shaped thermal material near a QCP, whose spacetime topology is $\mathbb{S}^1 \times \mathbb{R}$. In this case, the dual 3-dimensional gravity theory is defined on the region surrounded by the ring.

The experimental verification of SEMs will help us understand the conjecture and unveil the mystery of quantum gravity. To do so, we first reconstruct the bulk from limited experimental data of the candidate material, and check if the reconstructed bulk is capable of predicting other phenomena and consistent with the succeeding experiments. Therefore, as a first step, we need to establish a universal way to determine the bulk metric from available experimental data.

## 3. Method

In this study, we consider a theoretical setup for a material experiment of acting a small external source to some local operator in the theory and measuring its linear response function. Thus the boundary data available for us is the values of the source and the response, which according to the AdS/CFT dictionary are related to the asymptotic behavior of a scalar field in a gravitational curved spacetime. To reconstruct the bulk metric, we use the NN based equivalent to the Klein-Gordon equation of the scalar field on the unknown metric.

Let $\Phi(t, \theta, \xi)$ be the scalar field in the 3-dimensional bulk, where $t$ is the time coordinate, $\theta$ is the coordinate along $\mathbb{S}^1$, and $\xi \in [0, 1]$ is the radial coordinate (the extra dimension unique to the bulk). It is always possible to choose $\xi$ so that $\xi = 1$ corresponds to the ring on which the dual CFT lives. We Fourier-expand the scalar field as $\Phi(t, \theta, \xi) = \sum_n \Phi_n(\xi)e^{-i\omega t + ik_n\theta}$, where $k_n = 2\pi n/a$ with $a$ being the ring circumference, and $\omega$ is the frequency of the external source that is controllable in experiment. Since our interest is the forced oscillation part which remains after enough time has passed, only the component with frequency $\omega$ is taken into consideration. We introduce the conjugate of the scalar field $\Pi_n(\xi) = \Phi'_n(\xi)$, with which the Klein-Gordon equation is reduced to the first order equations. This is essential for the equation of motion to be rewritten as a NN architecture.

In the phase of verification of SEM, it is reasonable to use a material that is static, rotationally symmetric, and in

equilibrium. Supposing this situation, we can assume that the metric is of the following form:

$$ds^2 = -g_{tt}(\xi)dt^2 + g_{\xi\xi}(\xi)d\xi^2 + g_{\theta\theta}(\xi)d\theta^2. \quad (1)$$

The Klein-Gordon equation with this metric is

$$0 = -\omega^2\Xi(\xi)\Phi_n(\xi) + \partial_\xi\left(\ln\sqrt{-g}g^{\xi\xi}\right)\Pi_n(\xi)$$
$$+ \Pi'_n(\xi) - k_n^2\Theta(\xi)\Phi_n(\xi). \quad (2)$$

Here we have introduced the product combinations of the metric:

$$\Xi(\xi) = g_{\xi\xi}g^{tt}, \quad \Theta(\xi) = g_{\xi\xi}g^{\theta\theta}. \quad (3)$$

By using the residual redundancy of the diffeomorphism, we gauge-fix the metric as

$$\sqrt{-g}g^{\xi\xi} = C\xi^{-1}, \quad (4)$$

where any constant is allowed for $C$ and we will choose a value for it later. The validity of this gauge-fixing is explained in appendix A, where we will also see how to recover the metric under this gauge condition. Thus, all we have to do to reconstruct the metric is to determine $(\Xi, \Theta)$.

## 3.1. Numerical accuracy and discretization scheme

Discretizing (2), we would like to provide a NN representation of it. Before doing so, we have to determine the solver of the EOM, since the discretization scheme depends on the solver. Here we compare two solvers: the Euler method and the Runge-Kutta method. We use these solvers to numerically solve the EOM (2) on the BTZ black hole metric respectively, and compare how these solvers reproduce the exact solution to find which solver is suitable for our machine learning in later sections.

### 3.1.1. EXACT SOLUTION

On the BTZ black hole metric:

$$ds^2_{\text{BTZ}} =$$
$$-\frac{r_h^2\xi}{L^2(1-\xi)}dt^2 + \frac{L^2}{4\xi(1-\xi)^2}d\xi^2 + \frac{r_h^2}{L^2(1-\xi)}d\theta^2,$$
$$(5)$$

the exact solution of the EOM (2) satisfying the in-going boundary condition on the horizon is given as

$$\Phi_n = \xi^{\frac{\alpha_n+\beta_n}{2}}F(\alpha_n, \beta_n, \gamma_n; \xi), \quad (6)$$

where F is a Hypergeometric function, and $\alpha_n$, $\beta_n$ and $\gamma_n$ are defined as

$$\alpha_n := -i\left(\frac{L^2}{2r_h}(\omega + k_n)\right), \quad \beta_n := -i\left(\frac{L^2}{2r_h}(\omega - k_n)\right),$$
$$\gamma_n := 1 + \alpha_n + \beta_n. \quad (7)$$

In this expression, $L$ and $r_h$ are respectively related to the cosmological constant $\Lambda$ and the system temperature $T$ as

$$\Lambda = -\frac{1}{L^2}, \quad r_h = 2\pi L^2 T. \quad (8)$$

The temperature $T$ is controllable in the experiment. We set $L = r_h = 1.00$ for the numerical analysis below, and $C = 2r_h^2/L^2$ in (4) for the simplicity of the comparison between the true value of $(\Xi, \Theta)$ and the learned one.

### 3.1.2. EULER METHOD

Euler method uses the following recurrence relation to determine the values of $\Phi_n$ and $\Pi_n$:

$$\mathbf{Z}_n(\xi + \Delta\xi) =$$
$$\begin{pmatrix} 1 & \Delta\xi \\ \Delta\xi(\omega^2\Xi(\xi) + k_n^2\Theta(\xi)) & 1 - \Delta\xi/\xi \end{pmatrix}\mathbf{Z}_n(\xi). \quad (9)$$

Here, $\mathbf{Z}_n = (\Phi_n, \Pi_n)^{\text{T}}$ and $\Delta\xi$ is the discretization unit (the lattice constant) in $\xi$, which in this study is set to be equal to $-10^{-2}$. The initial value of $\mathbf{Z}_n$ is set to the value of the exact solution at $\xi = 0.99$ and we use the recurrence relation repeatedly to obtain $\mathbf{Z}_n(\xi = 0.1)$.

### 3.1.3. RUNGE-KUTTA METHOD

To illustrate the Runge-Kutta method, we first rewrite the EOM (2) into a vector form:

$$\mathbf{Z}'_n(\xi) = \mathbf{F}(\xi, \mathbf{Z}_n(\xi)), \quad (10)$$
$$\mathbf{F}(\xi, \mathbf{Z}) = \begin{pmatrix} 0 & 1 \\ \omega^2\Xi(\xi) + k_n^2\Theta(\xi) & -1/\xi \end{pmatrix}\mathbf{Z}. \quad (11)$$

The recurrence relation of the Runge-Kutta method is

$$\mathbf{Z}_n(\xi + \Delta\xi) = \mathbf{Z}_n(\xi) + \frac{\Delta\xi(\mathbf{F}_1 + 2\mathbf{F}_2 + 2\mathbf{F}_3 + \mathbf{F}_4)}{6}, \quad (12)$$

with

$$\mathbf{F}_1 = \mathbf{F}(\xi, \mathbf{Z}_n(\xi)),$$
$$\mathbf{F}_2 = \mathbf{F}\left(\xi + \frac{\Delta\xi}{2}, \mathbf{Z}_n(\xi) + \mathbf{F}_1\frac{\Delta\xi}{2}\right),$$
$$\mathbf{F}_3 = \mathbf{F}\left(\xi + \frac{\Delta\xi}{2}, \mathbf{Z}_n(\xi) + \mathbf{F}_2\frac{\Delta\xi}{2}\right),$$
$$\mathbf{F}_4 = \mathbf{F}(\xi + \Delta\xi, \mathbf{Z}_n(\xi) + \mathbf{F}_3\Delta\xi).$$

We use the same initial value as the Euler Method and find $\mathbf{Z}_n$ for $\xi \in [0.1, 0.99]$.

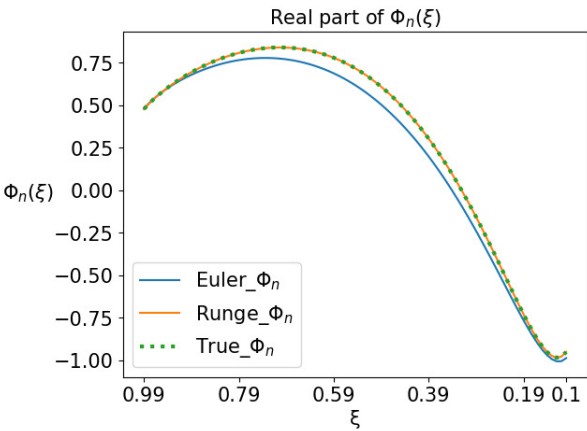
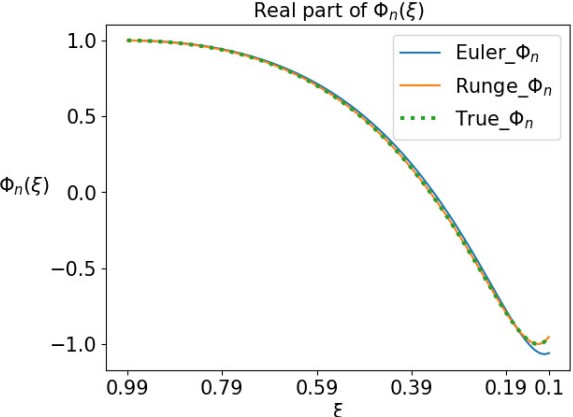

*Figure 1.* The profile of $\mathrm{Re}\Phi_n(\xi)$ for the exact solution, and numerical results with the Euler method and the Runge-Kutta method. For the parameters, we have chosen $(k_n, \omega) = (3.00, 3.00)$ (left) and $(k_n, \omega) = (2.00, 3.00)$ (right), respectively. The blue line, orange line, and green dotted line represent the Euler method, the Runge-Kutta method, and the exact solution respectively.

### 3.1.4. COMPARISON

We compare the above two solvers by examining their numerical reproduction accuracy against the exact solution. Fig.1 shows the numerical calculation results for the real part of $\Phi_n$. As can be seen from this figure, the Runge-Kutta method better approximates the exact solution than the Euler method. In fact, for the real component of $\Phi_n$ at $\xi = 0.1$, the relative error between the Runge-Kutta result and the exact solution is $1.5 \times 10^{-4}$ while the one between the Euler result and the exact solution is $7.5 \times 10^{-2}$, for the case of $k_n = \omega = 1.00$.

This result apparently shows that the Euler method used in (Hashimoto et al., 2018a) is not sufficient for numerical reconstruction of the differential equation for the present case.[2] Thus, as we will describe in the next subsection, we construct a NN model based on the Runge-Kutta method.

### 3.2. Neural network model based on Runge-Kutta method

Based on the results above, we choose the Runge-Kutta method and construct the NN. The structure of our NN model is shown in Fig. 2. We use the Runge-Kutta layer multiple times as well as the single boundary condition layer, where both layers are illustrated in the succeeding subsections.

---

[2]Note that the data structure and the bulk equations of motion in the present study are different from those in (Hashimoto et al., 2018a): the latter uses the data only at $k_n = \omega = 0$, and uses a nonlinear differential equation. So, a naive comparison with the present work using (Hashimoto et al., 2018a) as a benchmark is not possible.

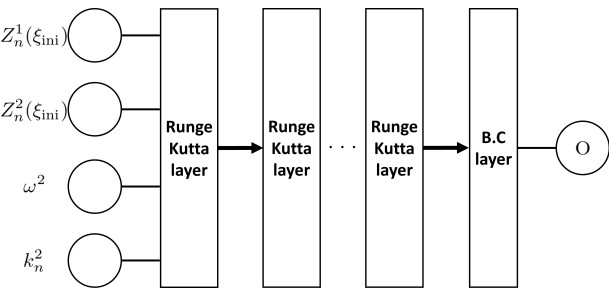

*Figure 2.* The model architecture.

### 3.2.1. RUNGE-KUTTA LAYER

The implementation of the Runge-Kutta layer is given by (12), where the input and the output of this layer are $(\mathbf{Z}_n(\xi), \omega^2, k_n^2)$ and $(\mathbf{Z}_n(\xi + \Delta\xi), \omega^2, k_n^2)$. The flow of data in the Runge-Kutta layer is depicted in fig. 3. The Runge-Kutta layer involves four custom layers, bulk layers to compute $\mathbf{F}_1, \mathbf{F}_2, \mathbf{F}_3, \mathbf{F}_4$ in turn. Fig. 4 shows the structure of the bulk layer. The bulk layer receives a four dimensional vector $(\mathbf{Z}, \omega^2, k_n^2)$ and returns a two dimensional vector $\mathbf{F}(\xi, \mathbf{Z})$, where $\mathbf{F}(\xi, \mathbf{Z})$ is defined in (11).

In this study, the numerical range of $\xi$ in the radial direction is set to $0.99 \sim 0.10$, and the discretization is performed with $\Delta\xi$, the incremental width, set to $-10^{-2}$. Therefore, the number of Runge-Kutta layers is 89.

### 3.2.2. BOUNDARY CONDITION (BC) LAYER

The gravity spacetimes can take two different topologies: one is the topology the BTZ black hole, and the other is

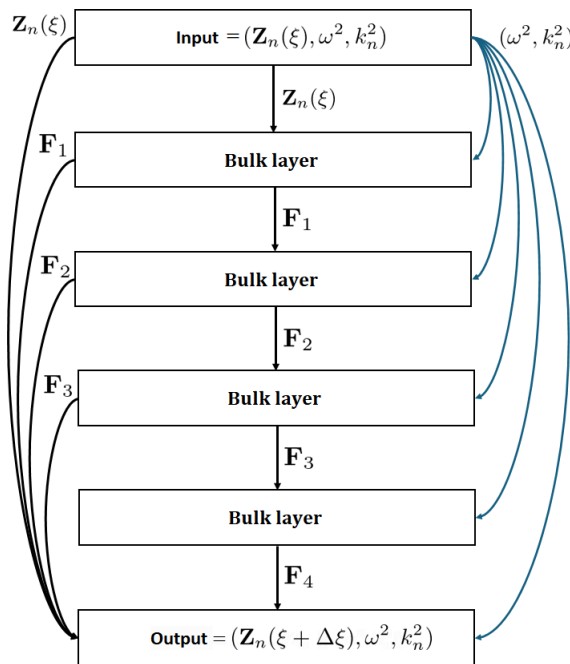

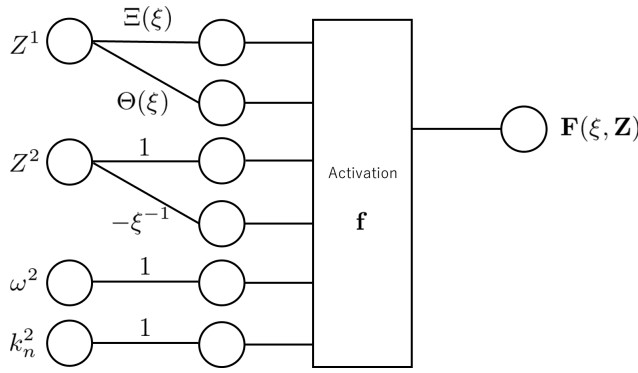

*Figure 4.* The bulk layer. Here $Z^{1,2}$ is a component of $\mathbf{Z}$ and the activation $\mathbf{f}$ is a function of a six dimensional vector $\mathbf{x} = (x_1, x_2, \ldots, x_6)$ and gives the four dimensional vector $(x_3, x_4 + x_5 x_1 + x_6 x_2, x_5, x_6)$.

*Figure 3.* The data flow of the Runge-Kutta layer.

that the AdS soliton. One of these is supposed to be favored depending on the temperature. These topologies have different boundary conditions at the deepest part of the radial direction. Therefore, in order to reconstruct the boundary conditions with learning, we introduce the complex trainable parameter $\rho_n$ which can control the boundary conditions, at the deepest layer of the neural network, as follows:

$$\frac{\xi_{\text{fin}} \Pi_n(\xi_{\text{fin}}) + \rho_n \Phi_n(\xi_{\text{fin}})}{\sqrt{|\xi_{\text{fin}} \Pi_n(\xi_{\text{fin}})|^2 + |\rho_n \Phi_n(\xi_{\text{fin}})|^2 + \epsilon}} = 0. \quad (13)$$

Here $\Phi_n(\xi_{\text{fin}})$, $\Pi_n(\xi_{\text{fin}})$ are the values of the fields at $\xi_{\text{fin}}$. $\epsilon$ is the regularization parameter such that the expression (13) is numerically well-defined, In practice, we take $\epsilon = 10^{-6}$. To be more precise, $\rho_n$ is parameterized as

$$\rho_n = i\omega a_n + |k_n| b_n, \quad (14)$$

with real parameters $a_n$ and $b_n$.

This form incorporates both of the boundary conditions for the BTZ black hole and the AdS soliton, with the following choices for $\rho_n$:

$$a_n = \frac{L^2}{2r_h}, b_n = 0, \quad \text{for BTZ}, \quad (15)$$

$$a_n = 0, b_n = \frac{L^2}{2r_s}, \quad \text{for AdS soliton}. \quad (16)$$

For example, the above choice of $a_n, b_n$ for the BTZ case implies the in-going boundary condition. In summary, the

trainable parameters of our NN model are $\Xi(\xi), \Theta(\xi)$ and a complex parameter $\rho_n$.

### 3.2.3. INITIAL WEIGHTS

Since our purpose is to reconstruct an asymptotically AdS spacetime[3], it is reasonable to choose the initial weights as those of the pure $\text{AdS}_3$ spacetime:

$$ds^2_{\text{AdS}_3} = -\frac{1}{1-\xi}dt^2 + \frac{1}{4\xi(1-\xi)^2}d\xi^2 + \frac{\xi}{1-\xi}d\theta^2. \quad (17)$$

In Fig. 5, we show the profiles of the pure AdS spacetime, $(\Xi, \Theta)$. Moreover, the initial values of the learning parameters $a_n, b_n$ included in the BC layer are set to $\frac{L^2}{2r_h}, \frac{L^2}{2r_s}$ respectively. In practice, we set $r_s = 1.0$ as well as $r_h$ and $L$.

### 3.2.4. LOSS FUNCTION

This study is a supervised learning, and the data has a label 0 or 1 according to whether it satisfies the boundary conditions at the deepest boundary of the spacetime. To implement

---

[3]"Asymptotically AdS" means that the metric approaches to (17) in $\xi \to 1$. In AdS/CFT correspondence, this is assumed most of the time.

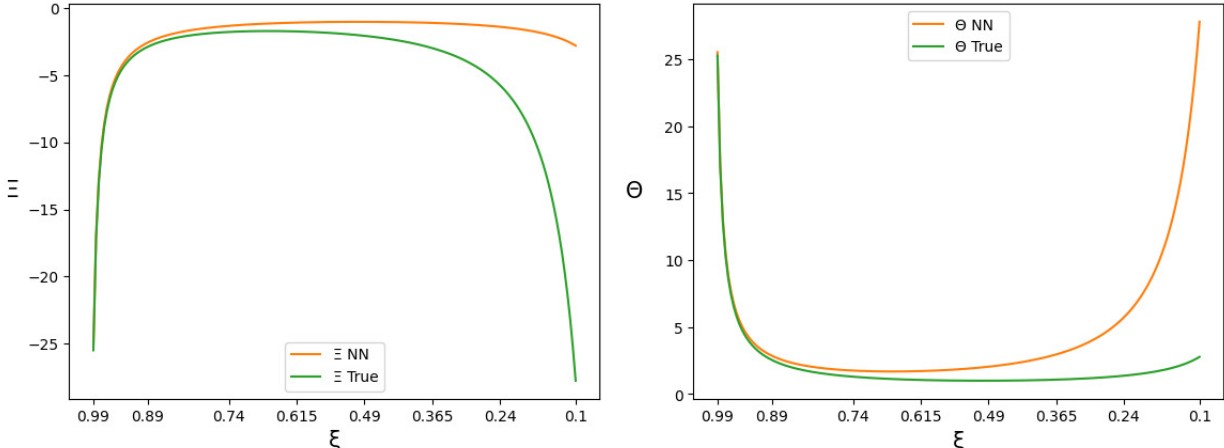

*Figure 5.* Profiles of the metric $(\Xi, \Theta)$ are shown. The orange lines represent the pure $\text{AdS}_3$ metric (17) which is the initial condition. The green lines represent the BTZ black hole metric (1) which is used for generating the training set and also is expected to be reproduced by the learning.

this, we adopt the following loss function $L$ for the training,

$$L(t) = -t_{\text{data}} \log t - (1 - t_{\text{data}} + \epsilon) \log(1 - t + \epsilon),$$

$$(18)$$

$$t = \frac{1}{2}\left[\tanh\left(100(\text{O} - 0.1)\right) - \tanh\left(100(\text{O} + 0.1)\right)\right],$$

$$(19)$$

$$\text{O} = \left| \frac{\xi_{\text{fin}}\Pi_n(\xi_{\text{fin}}) + \rho_n\Phi_n(\xi_{\text{fin}})}{\sqrt{|\xi_{\text{fin}}\Pi_n(\xi_{\text{fin}})|^2 + |\rho_n\Phi_n(\xi_{\text{fin}})|^2 + \epsilon}} \right|, \qquad (20)$$

where O is the output of the BC layer and is the LHS of (13). Notice that $\Phi_n(\xi_{\text{fin}})$ and $\Pi_n(\xi_{\text{fin}})$ are obtained from the output of the last Runge-Kutta layer and $\rho_n$ is parameterized as (14). $t_{\text{data}}$ is the ground truth label and its determination method is shown in the subsequent subsection. As a result of (19), $t$ is close to 0 if O is sufficiently small but it suddenly approaches 1 if O moves away from 0.

### 3.3. Dataset

As we shall explain in this subsection, the element of our dataset consists of $\mathbf{Z}_n(\xi_{\text{ini}}), \omega^2, k_n^2, t_{\text{data}}$. Here we describe how to determine the value of $t_{\text{data}}$. In this study, we will demonstrate if the learning of our NN model reproduces the BTZ black hole metric in the higher temperature phase. The exact solution to the EOM (2) with the background of the BTZ black hole is a linear combination of two independent solutions (Hashimoto et al., 2023):

$$\Phi_n(\xi) = C_n^1 \xi^{\frac{\alpha_n + \beta_n}{2}} F(\alpha_n, \beta_n, \gamma_n; \xi)$$
$$+ C_n^2 \xi^{-\frac{\alpha_n + \beta_n}{2}} F(-\beta_n, -1 - \alpha_n, 1 - \alpha_n - \beta_n; \xi).$$

$$(21)$$

Here $C_n^1$ (or $C_n^2$) is a coefficient of in-going (or out-going) solution. When we expand this solution around the boundary ($\xi = 1$), we have

$$\Phi_n(\xi) \sim D_n^1(1 + \alpha_n\beta_n(1 - \xi)\ln(1 - \xi)$$
$$- (1 + \alpha_n\beta_n)(1 - \xi)) + D_n^2(1 - \xi)/r_h^2,$$
$$\Pi_n(\xi) \sim D_n^1(1 - \alpha_n\beta_n\ln(1 - \xi)) - D_n^2/r_h^2, \qquad (22)$$

with

$$D_n^1 = (C_n^2\Gamma(1 - \alpha_n - \beta_n))/(\Gamma(1 - \alpha_n)\Gamma(2 - \beta_n)))$$
$$+ ((C_n^1\Gamma(1 + \alpha_n + \beta_n))/(\Gamma(1 + \alpha_n)\Gamma(1 + \beta_n)),$$
$$D_n^2 = \frac{1}{2}r^2\frac{C_n^2(2 + \alpha_n - \beta_n)\Gamma(1 - \alpha_n - \beta_n)}{\Gamma(1 - \alpha_n)\Gamma(2 - \beta_n)}$$
$$+ C_n^1\Gamma(1 + \alpha_n + \beta_n)\{2 - \alpha_n - \beta_n + 4\gamma\alpha_n\beta_n$$
$$+ 2\alpha_n\beta_n\psi(1 + \alpha_n)$$
$$+ 2\alpha_n\beta_n\psi(1 + \beta_n)/\Gamma(1 + \alpha_n)\Gamma(1 + \beta_n)\}. \quad (23)$$

Here $\gamma$ is the Euler's constant and $\psi$ is the digamma function. According to the AdS/CFT dictionary, $D_n^1$ is regarded as the source of the force exerted on the SEM, and $D_n^2$ is the response of it.

Furthermore, for this to be exactly a solution in all regions of the spacetime, the solution needs to satisfy the boundary condition at the black hole horizon, which is the in-going boundary condition. So, the out-going term of this solution needs to be infinitesimally small: $C_n^2 = 0$.

Taking the above into consideration, the following steps are used to generate the dataset.

1. Generate $C_n^{1,2}$ randomly. We randomly sample the

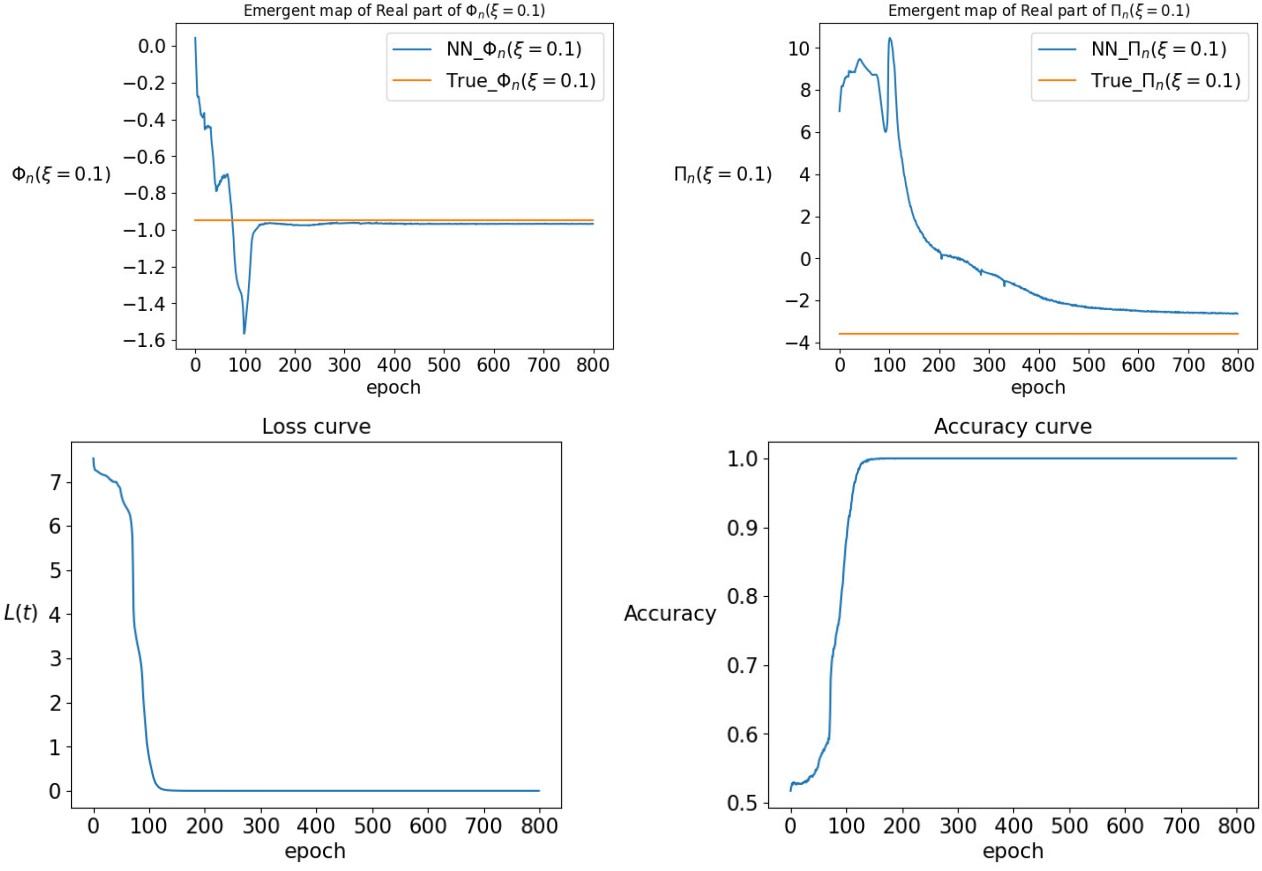

*Figure 6.* Top: Evolution of the scalar field values during the training. The left (right) panel is the value of the real part of $\Phi$ ($\Pi$) at $\xi = 0.1$ for each epoch of the training. The boundary value of the field is chosen to be that of the exact solution for $\omega = 3.0, k_n = 1.0$. Bottom: the loss curve and the accuracy curve during the training.

real and imaginary parts of $C_n^1$ from the uniform distributions $\mathcal{U}(0.01, 1.0)$ respectively. We adopt $C_n^1$ when $0.5 < |C_n^1| < 1.2$ otherwise we abandon it. On the one hand, the real and imaginary parts of $C_n^2$ are randomly sampled from $\mathcal{U}(0.001, 0.5)$.

2. Randomly drown $\omega, k_n$ from the uniform distribution $\mathcal{U}(0.0, 3.0)$ and $\mathcal{U}(0.0, \omega)$ respectively.[4]

3. Classify the positive/negative data by the following decision condition,

$$t_{\text{data}} = \begin{cases} 0 & (|C_n^2/C_n^1| < 0.01) : \text{positive} \\ 1 & (|C_n^2/C_n^1| > 1.00) : \text{negative} \end{cases} . \quad (24)$$

Here we discard the data if $0.01 \leq |C_n^2/C_n^1| \leq 1.0$.

4. Obtain $D_n^{1,2}$ from $C_n^{1,2}$ using equation (23).

---

[4]When $k_n > \omega$, the numerical solution of the scalar field on BTZ metric turns out to have a significant difference from the exact solution. To avoid this, $k_n$ is set to be smaller than $\omega$.

5. Obtain $\mathbf{Z}_n(\xi_{\text{ini}})$ through (22) by substituting $\xi_{\text{ini}}$ for $\xi$.

6. Formulate the data as $\{(\mathbf{Z}_n(\xi_{\text{ini}}), \omega^2, k_n^2, t_{\text{data}})\}$.

We repeat this procedure until we obtain 1000 examples for $t_{\text{data}} = 0$ and $t_{\text{data}} = 1$ respectively to obtain 2000 examples in total.

## 4. Results

We build a Neural Network based on the Runge-Kutta method. The following hyperparameters are commonly used in this study: optimizer = Adam, batch size = 10, epoch = 800.

Before training, the class loss was 7.75. After 800 epochs of training, the class loss value decreased to $-1.02 \times 10^{-7}$, with an accuracy of 1.0. The loss went down significantly, and the training was successful. See Fig. 6.

In addition, in this study, the parameter $(a_n, b_n)$ in the ex-

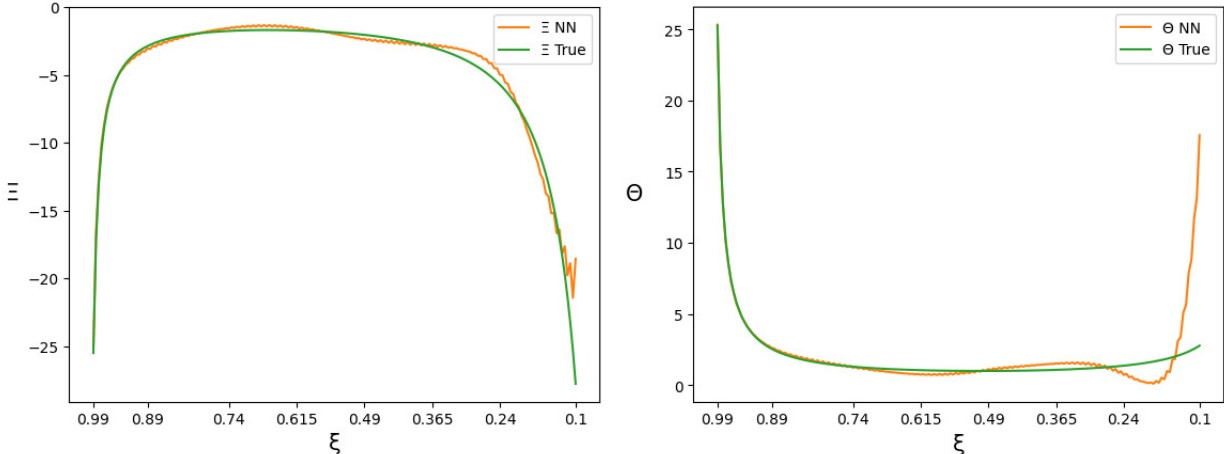

*Figure 7.* Profiles of $\Xi, \Theta$ after learning. The orange line is the weights of the NN, and the green line is the ground truth, the BTZ metric (1).

pression for the boundary condition was introduced as a learning parameter to discriminate spacetimes at the BC layer. After the learning, this parameter $(a_n, b_n)$ is found to take a value $(0.50, 0.01)$. If the spacetime is a BTZ black hole, the solution of equation (15) gives the value of $(a_n, b_n)$ equal to $(0.50, 0.00)$, which means that we obtained the ingoing boundary condition at the black hole horizon properly: *The emergence of the black hole horizon is successful.*

Fig. 7 shows profiles of metric $\Xi, \Theta$ after learning. Compared to the initial condition shown by the orange lines in Fig. 5, the tendency of the correct BTZ metric is finely reproduced, and the metric is smooth almost everywhere so that we can interpret it as a smooth geometry emergent.

A finer look at the near horizon part of the emergent geometry $\xi \sim 0.1$, we find that the profile of $\Theta$ shown in Fig. 7 appears to be a little bit deviated from the ground truth. However, this is expected, as this $\Theta$ is related to the angular direction of the metric whose deep IR part near the horizon is largely affected by black hole red shift, meaning that high $\omega$ and large $k_n$ data is needed to probe this part of $\Theta$. Since we used only the range $|\omega|, |k_n| \lesssim \mathcal{O}(3)$, the difficulty in a fine reproduction of $\Theta$ near the horizon was expected.

## 5. Future Directions

When using a neural network to extract the information of a dual gravitational spacetime from the SEM response function, the radial direction $\xi$ must be discretized, and numerical results show that the accuracy is insufficient unless the Runge-Kutta method is used. Therefore, we construct a neural network model based on the Runge-Kutta method

and propose a reconstruction of BTZ black hole spacetime by using the coefficients in the equation as training parameters. As a result, we succeeded in reconstructing the BTZ black hole.

Nevertheless, this study still has room for improvement. For example, in this study, we built a layer called BC Layer, which identifies the spacetime based on the satisfaction of boundary conditions. Although it can be said that the BTZ black hole was successfully identified after training, it cannot be said that it can identify any spacetime, since it has not yet identified other spacetimes such as the AdS soliton. Therefore, we plan to reconstruct the spacetime of AdS soliton from the response function of a SEM.

One concern is about the initial condition. We used the pure AdS spacetime for the initial condition, and we are interested in whether the BTZ black hole is reconstructed even with some other initial condition for the weights. In Appendix B, we discuss different initial weights.

Another concern is the question about whether our neural network architecture is not too sparse to obtain the bulk spacetimes. In fact, in the metric we have two unidentified functions $(\Xi(\xi), \Theta(\xi))$ which are to be determined by learning, while the data is a single linear response function on the ring against a source which has a temporal and spatial (angular) dependence in general, $J(\omega, k_n)$. So, regarding the dimensionality, our system is over-deterministic. However, since we used the data generated only in a finite region of $k_n$, and also since the number of data points is finite, the dimensionality argument based on continuous functions may not be sufficient. In fact, to improve the metric function near the horizon, physically it is expected to be necessary to

have the data with large $k_n$ and $\omega$.[5]

For these reasons, this study may be extended to be tested in more variety of situations. If this research makes it possible to reconstruct any spacetime, it will be possible to combine physics and machine learning to conduct quantum gravity experiments that were previously impossible.

## 6. Broader Impact

In general, in scientific application of machine learning, interpretability is indispensable. Although one could obtain some solutions of a given equation, if the numerical solution does not allow any good human interpretation, we cannot expect any scientific progress from the obtained numerical solutions. Here in this work, we express the differential equation with unknown functional coefficient in terms of a novel neural network, and train it by using the data of the boundary information of its solutions. With an appropriate physical intuition for the sparsity of the neural network, the training was done successfully and the emergent network allows a spacetime interpretation. In this way, the inverse problem of reconstructing the differential equation is done by the machinery of AI, by imposing a physical bias appropriately.

This demonstrates that in any physically interpretable AI one needs to impose an appropriate physical conditions on the AI architecture itself. Of course, one could use some physical regression from the obtained NN weights, but normally the dimensions of the NN weight space is huge, it is almost impossible. Neural ODE (Chen et al., 2018) is not an exception. In view of this, sparse neural network such as recently proposed KANs (Kolmogolov-Arnold networks) (Liu et al., 2024) could be used for future study. Our study is solely based on the most popular feed-forward neural network with sparsity imposed, and this could be thought of as a firm example exhibiting the effectiveness of the imposition of the sparsity based on physical knowledge.

Study of AI for inverse problems is not restricted to the interpretable architecture which we studied in detail; Transformers(Vaswani et al., 2017) may be combined to execute mathematical regressions (Kamienny et al., 2022; Charton, 2021) based on the correspondence between mathematical expressions and language sentences (Lample & Charton, 2019). In fact, recently Transformers were used for particle theory problems (Park et al., 2023; Cai et al., 2024; Hashimoto et al., 2024), among which the application of Transformers to decoding scattering amplitudes in $\mathcal{N} = 4$

Super Yang-Mills theory (Cai et al., 2024) has a quite close subject to ours, as that theory is the most popular model in AdS/CFT correspondence. We expect that the combination of the present formulation and other machine learning techniques will enhance the solving ability of inverse problems by AI.

## Acknowledgements

K. H. and D. T. would like to thank Keiju Murata for valuable discussions. The work of K. H., K. M., M. M. and G. O. was supported in part by JSPS KAKENHI Grant Nos. JP22H01217, JP22H05111, and JP22H05115. The work of D. T. was supported in part by Grant-in-Aid for JSPS Fellows No. 22KJ1944.

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

## A. On the gauge-fixing

Here, we will explain that the gauge condition (4) is possible.

First, we find the generic behavior of $\sqrt{-g}g^{\xi\xi}$ in $\xi \to 1$. Since the metric is asymptotically AdS, the asymptotic form of any such metric is generated by using a coordinate transformation of (17). As the metric is assumed to be static and rotationally-symmetric in (1), the remaining gauge freedom is limited to the transformation as $\eta = \eta(\xi)$, where $\eta$ is a smooth bijection map $[0, 1] \to [0, 1]$. From (17), the asymptotic form of the metric in the $\eta$ coordinate will be

$$ds^2 \sim -\frac{\alpha}{1 - \xi(\eta)}dt^2 + \frac{\xi'(\eta)^2}{4\xi(\eta)(1 - \xi(\eta))^2}d\eta^2 + \frac{\alpha\,\xi(\eta)}{1 - \xi(\eta)}d\theta^2 \qquad (\eta \to 1), \tag{25}$$

with $\xi(\eta)$ being the inverse mapping of $\eta(\xi)$. Here, we have taken into account not only the gauge-transformation, but also the possibility of an extra constant factor $\alpha$ as in (5). In this coordinate, we have

$$(\sqrt{-g}g^{\eta\eta})^{-1} \sim \frac{\alpha}{2}\xi'(\eta) \qquad (\eta \to 1). \tag{26}$$

Then, the r.h.s. could diverge depending on the gauge choice, but the integral is always convergent at $\eta \to 1$ owing to $\xi(1) = 1$.

Thus, from the above, we conclude

$$\int^1 d\xi\, (\sqrt{-g}g^{\xi\xi})^{-1} = \int^1 d\xi\, \sqrt{\frac{g(\xi)}{f(\xi)h(\xi)}} = \text{finite} \tag{27}$$

for general metric (1) unless the lower limit is a singularity. This fact is essential for guaranteeing that our gauge condition is possible.

Next, we survey the limit $\xi \to 0$. The behavior of the metric in this limit differs depending on whether the spacetime contains a black hole or not. When there is no black hole, it usually happens that the $\mathbb{S}^1$ of $t = \text{const}$ and $\xi = \text{const}$ in (1) shrinks to vanish as $\xi \to 0$, while $f(0)$ takes finite nonzero value. Thus, we can write $h(\xi) \sim A^2\xi^{2n}$ with some positive constants $A$ and $n$.

To avoid conical singularity, $(\xi, \theta)$-plane must approach to the flat $\mathbb{R}^2$ in $\xi \to 0$. Performing a coordinate transformation $r = \xi^n$, we have

$$g(\xi)d\xi^2 + h(\xi)d\theta^2 \sim \frac{g(r^{1/n})r^{2/n-2}}{n^2}dr^2 + A^2r^2d\theta^2 \qquad (\xi \to 0), \tag{28}$$

where $a$ is the periodicity of $\theta$. For the r.h.s. to be flat, $g(\xi)$ must behave as

$$g(\xi) \sim \left(\frac{Aan}{2\pi}\right)^2 \xi^{2(n-1)} \qquad (\xi \to 0). \tag{29}$$

Then, there exists a positive constant $B$ such that

$$\sqrt{-g}g^{\xi\xi} = \sqrt{\frac{f(\xi)h(\xi)}{g(\xi)}} \sim B\xi \qquad (\xi \to 0). \tag{30}$$

When there is a black hole, the same result is reproduced with the roles of $f$ and $h$ exchanged. In our convention of $\xi$, $\xi = 0$ is supposed to be the horizon, where $f$ vanishes and is written of the form $f \sim A^2\xi^{2n}$.[6] Recalling that the periodicity of $t$ in the Euclidean version of (1) is given by $T^{-1}$, the same logic as above again leads to (29) with the replacement $a \to T^{-1}$. Thus, we obtain (30) in this case as well.

---

[6]Technically speaking, the horizon of a static black hole is characterized as the hypersurface on which the Killing vector $\partial_t$ becomes null. This is equivalent to $f(0) = 0$.

Now, our task is to find a smooth bijective map $\phi : [0,1] \to [0,1]$, $\phi = \phi(\xi)$, with $\phi(0) = 0$ and $\phi(1) = 1$ such that the following is satisfied in the $\phi$ coordinate:

$$\sqrt{-g}g^{\phi\phi} = \sqrt{\frac{f(\xi(\phi))h(\xi(\phi))}{g(\xi(\phi))}} \frac{1}{\xi'(\phi)} = C\phi. \tag{31}$$

Here, $C$ must be positive since the l.h.s. is positive, and $f$, $g$, and $h$ are those in (1), which is a generic asymptotically AdS metric that satisfies (27) and (30). Seeing (31) as a differential equation, we have

$$\phi(\xi) = \exp\left[C \int_1^\xi d\xi' \sqrt{\frac{g(\xi')}{f(\xi')h(\xi')}}\right]. \tag{32}$$

By (27), the integrand in the r.h.s. is well-defined. In the limit $\xi = \epsilon \ll 1$, we obtain from (30)

$$\phi(\epsilon) \sim \exp\left[\frac{C}{B} \int^\epsilon d\xi \frac{d\xi'}{\xi'}\right] \sim \epsilon^{C/B}, \tag{33}$$

which is consistent with $\phi(0) = 0$. We also see that the r.h.s. of (32) is in $[0,1]$ and monotonically increasing for $\xi \in [0,1]$, because the integrand is always positive. Therefore, (32) concretely provides a smooth bijection map $\phi$ with $\phi(0) = 0$ and $\phi(1) = 1$, which is the transformation to reproduce (31).

Under the gauge condition (4), all the metric components are recovered form $(\Xi, \Theta)$ as

$$f(\xi) = -g_{tt}(\xi) = C\xi^2\Theta, \qquad g(\xi) = g_{\xi\xi}(\xi) = -C\xi^2\Xi\Theta, \qquad h(\xi) = g_{\theta\theta}(\xi) = -C\xi^2\Xi. \tag{34}$$

## B. Initial weight dependence

As we stated, the initial condition for the geometry used in the neural network is the pure AdS spacetime (17). In this appendix, we report the training results with a different initial condition to check whether our obtained result is general or not. Our choice for the different initial condition is just a constant function for $\Xi(\xi)$ and $\Theta(\xi)$. With precisely the same dataset, architecture and weight update methods, we obtain the trained weights which are shown in Fig. 8.

As we can see in Fig. 8, the obtained metric is consistent with the BTZ black hole metric, while the deviation from the ground truth is bigger compared to the case of the pure AdS initial condition reported in Fig. 7. The reason is obvious: Since the BTZ black hole metric is asymptotically AdS, meaning that at large $\xi$ it is equal to the pure AdS, the training should be easier for the pure AdS initial condition. Nevertheless, it is encouraging that, even with the constant initial condition, the trained metric is confirmed to be consistent with the BTZ black hole metric.

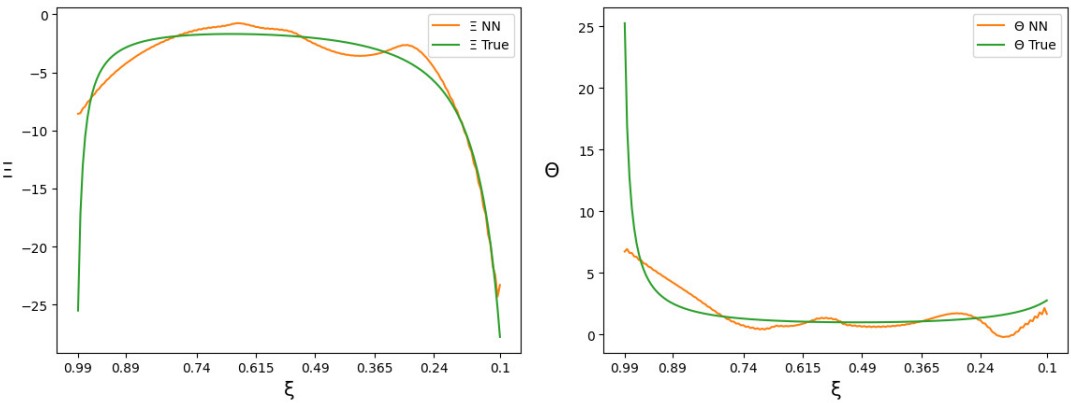

*Figure 8.* Profiles of $\Xi, \Theta$ after learning with the constant initial condition. The orange line is the weights of the NN, and the green line is the BTZ metric (5).

## C. Physics-informed regularization

In our future study we plan to use some actual experimental data of a ring-shaped material, and in that case, the emergent bulk metric will not be a solution of an Einstein equation. Furthermore, even for a single set of data, there could be various emergent spacetimes, depending on the amount of data.[7] To single out a plausible geometry which is physically easier to be interpreted, here we introduce the notion of free energy regularization to measure to what extent the emergent geometry satisfies the Einstein equation. As has been mentioned around (1), the system is supposed to be at a thermal equilibrium, which physically means that the free energy takes its minimum for a fixed temperature $T$.

In the AdS/CFT, the leading term of the free energy in the expansion of the string scale in the bulk is known to be the following Euclidean action[8]:

$$S_{\mathrm{E}} = S_{\mathrm{EH}} + S_{\mathrm{GH}} + S_{\mathrm{CT}},$$

$$S_{\mathrm{EH}} = 2\pi\beta \int_{\xi_{\mathrm{fin}}}^{\xi_{\mathrm{ini}}} d\xi \sqrt{fgh} \left[ \frac{1}{L^2} - \mathrm{Ricci} \right], \quad S_{\mathrm{GH}} = -2\pi\beta \sqrt{\frac{fh}{g}} \left( \frac{f'}{f'} + \frac{h'}{h} \right)\bigg|_{\xi_{\mathrm{ini}}}, \quad S_{\mathrm{CT}} = \frac{4\pi\beta}{L} \sqrt{fh} \bigg|_{\xi_{\mathrm{ini}}},$$

$$\mathrm{Ricci} = \frac{1}{2f^2 g^2 h^2}(fg'h(fh)' + f^2 gh'^2 + f'^2 gh^2 - 2ff''gh^2 - ff'ghh' - 2f^2 ghh''), \tag{35}$$

where $S_{\mathrm{EH}}$ is Einstein-Hilbert action, $\mathrm{S}_{\mathrm{GH}}$ is Gibbons-Hawking-York boundary term (Chakraborty, 2017), and we have introduced

$$f(\xi) = -g_{tt}(\xi) = -\frac{g_{\xi\xi}(\xi)}{\Xi(\xi)}, \qquad g(\xi) = g_{\xi\xi}(\xi) = -\frac{4r_h^4 \xi^2 \Xi(\xi)\Theta(\xi)}{L^6}, \qquad h(\xi) = g_{\theta\theta}(\xi) = \frac{g_{\xi\xi}(\xi)}{\Theta(\xi)}. \tag{36}$$

Since the action is Euclidean, we have used the Euclidean signature of (1). The minimization of the action is equivalent to the equations of motion which is the Einstein equation.

Then, to ensure that the system is in thermal equilibrium, we can introduce the following regularization term:

$$R_{\mathrm{E}} = S_{\mathrm{E}}. \tag{37}$$

To complete the free energy regularization, the temperature must be fixed. The temperature is in the bulk description provided by the metric value on the horizon as

$$T = \frac{1}{4\pi} \frac{f'(\xi_{\mathrm{fin}})}{\sqrt{f(\xi_{\mathrm{fin}})g(\xi_{\mathrm{fin}})}}. \tag{38}$$

This is derived by imposing the regularity of the Euclidean version of (1) at $\xi = 0$, which determines the inverse temperature $T^{-1}$, the periodicity of the Euclidean time. Thus, in addition to (37), we further have to introduce

$$R_{\mathrm{T}} = \left( T - \frac{1}{4\pi} \frac{f'(\xi_{\mathrm{fin}})}{\sqrt{f(\xi_{\mathrm{fin}})g(\xi_{\mathrm{fin}})}} \right)^2, \tag{39}$$

as a regularization term.

We train the NN with the addition of the regularization terms $R_{\mathrm{E}}$ and $R_{\mathrm{T}}$, and find that our previous results shown in Sec. 4 are not altered significantly, while the loss value of the newly added regularization terms are significantly small. This suggests that our obtained emergent metric is close to the BTZ black hole, as we stated in Sec. 4.

Here is a technical note on the training protocols. First we train the NN using the loss function (18). Then after 400 epochs, we add (37) and (39). This is because the initial condition (which is the pure AdS spacetime) is already one of the stationary points of the action $S_{\mathrm{E}}$. It was expected that our choice of the initial weight in (17) is trapped near the initial value in the weight space and prevents the NN from being successfully trained.

---

[7]For example, in our case, we have used limited range of $\omega$ and $k_n$. If one uses data with all possible values of $\omega$ and $k_n$, then one should be able to pin down a single emergent geometry. However, for limited amount of data, a variety of emergent geometries may be learned.

[8]This kind of "Einstein regularization" was described in (Hashimoto et al., 2018a) and (Hashimoto, 2019).

After $400$ epochs of the learning with the additional regularizations (37) and (39), the total loss is $1.16$, the class loss is $1.00 \times 10^{-7}$, the accuracy is $1.00$ and $(a_n, b_n)$ is $(0.50, 0.03)$. These results are comparable to the ones in the previous section, suggesting that the emergent geometry is interpreted as a BTZ black hole. See Fig. 9 for the trained profiles of the metric functions.

The claim above is merely suggestive, since we have the following concern. A subtlety about the temperature regularization term (39) is that it can always be satisfied only by tuning of a few of the near-horizon weights. In fact, careful comparison between the profiles shown in Figs. 7 and 9 indicates that this very local adjustment was going on in learning, and physically the regularization (39) may not have worked well. We plan to come back to this concern in the future.

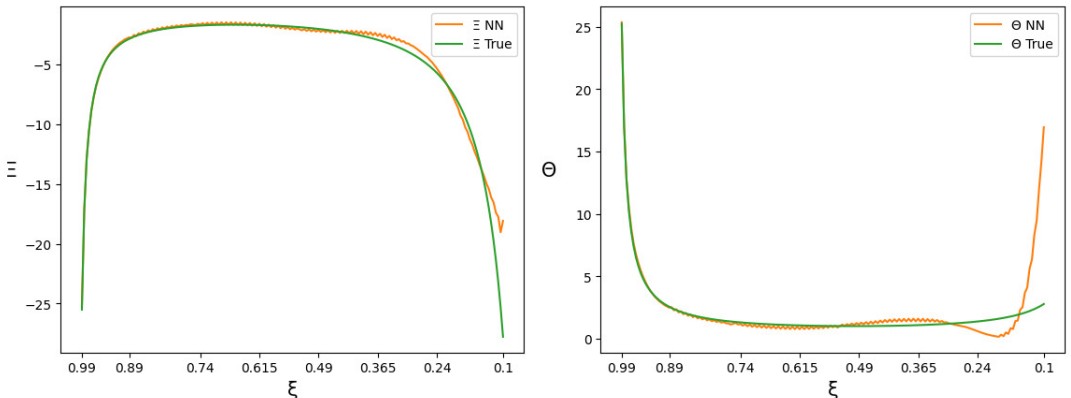

*Figure 9.* Profiles of $\Xi, \Theta$ after learning with the addition of the physics-informed regularization. The orange line is the weights of the NN, and the green line is the BTZ metric (5).

