# OpenReview forum: "AI for an inverse problem: Physical model solving quantum gravity"
_ICML.cc/2024/Workshop/AI4MATH — ICML 2024 Workshop AI4MATH Poster_

### Official Review · Reviewer_KLDC · 2024-06-11

**Rating:** 7
**Confidence:** 3

**Summary:**

This paper explores using Neural Networks to find solution to the inverse problem of the AdS/CFT correspondence. The author first demonstrated that Runge-Kutta method outperformed Euler method in solving EOM, and then incorporated a novel layer that implements the Runge-Kutta method.

**Questions:**

1. What is the time complexity of the proposed method?
2. Are the experimental results reproducible? Are you going to release the code that generates the experimental results?

**Reasons To Accept:**

1. This work is of broad interest since studying inverse problem is relevant to many fields.
2. This paper is well-written, and self-contained. Both theories and experimental results are well discussed.

**Reasons To Reject:**

1. It would be more convincing to add a benchmark method, e.g., the approach proposed in the work below

@article{hashimoto2018deep,
  title={Deep learning and the AdS/CFT correspondence},
  author={Hashimoto, Koji and Sugishita, Sotaro and Tanaka, Akinori and Tomiya, Akio},
  journal={Physical Review D},
  volume={98},
  number={4},
  pages={046019},
  year={2018},
  publisher={APS}
}

---

### Official Review · Reviewer_KDm6 · 2024-06-11

**Rating:** 6
**Confidence:** 2

**Summary:**

This paper articulates a novel interpretable implicit sparse neural network representation of the AdS/CFT correspondence. The authors train their model with the response functions of a condensed matter system, and the trained model is equivalent to a scalar field equation on an emergent geometry of the bulk spacetime. The proposed model also incorporates a novel layer that implements the Runge-Kutta method to achieve better numerical control. Acceptance is recommended since this work has enough novelty and presents sufficient preliminary experimental results.

**Questions:**

See "Reasons to Reject" above. I would like to see some additional experimental data and/or more details specifications on the experiments.

**Reasons To Accept:**

- The proposed method has sufficient novelty;
- This paper is in general well-written and the experimental results show enough preliminary results;
- The authors clearly report the current shortcomings of the proposed method and its potential extensions.

**Reasons To Reject:**

- The scope of experiments is a little bit limited (only AdS/CFT correspondence problems are modeled), and whether this framework could handle other possible Physical phenomena remains unclear;
- For AdS/CFT correspondence problem, the authors could try different parameter combinations to test the robustness of the proposed method.
- More details could be provided regarding the experiments, e.g., convergence maps during training, and visualization of the recovered scalar fields v.s. the ground truth value (like heat maps).

---

### Official Review · Reviewer_VSm5 · 2024-06-13

**Rating:** 5
**Confidence:** 2

**Summary:**

This paper tries to solve the AdS/CFT correspondence problem through deep learning. Specific model structures are designed for this problem.

**Questions:**

I wonder whether the designed layers are necessary. if possible, I suggest adding ablation studies on this.

**Reasons To Accept:**

This paper is well-written and easy to read. The proposed task is interesting.

**Reasons To Reject:**

Please see the questions below.

---

### Meta-Review · Area_Chair_kER6 · 2024-06-13

**Recommendation:** Accept (Poster)
**Confidence:** 4

**Metareview:**

Using AI techniques to suggest mathematical solutions to advanced problem in mathematics has been investigated for a few years now and is, of course, a very interesting question of AI and maths. The example of this paper is very interesting. Despite being only a preliminary result and a bet on the future, I believe this deserves publication. The addition of a Runge-Kutta layer is interesting. I agree with the two last comments of reviewer KDm6 (robustness, details on the experiments), and the questions of reviewer KLDC. I also agree with the comment of reviewer VSm5 but I think this should not be ground for rejection and is acceptable in a workshop. I would also encourage the author to look at the works of Charton for instance for different approaches on the same meta-question: AI for solving advanced maths problems (generally speaking 25% of the reference of this paper are the author's which may indicate a lack of literature review). I entrust the authors to make their best to answer the reviewers' comments that I highlight above in a potential camera-ready version. I recommend this paper for acceptation.

---

### Decision · Program_Chairs · 2024-06-13

Accept (Poster)